# EVA-FLOW: ENVIRONMENT-AWARE FLOW MATCHING FOR UNIFIED 3D MOLECULAR CONFORMATION GENERATION

## ABSTRACT

Predicting the 3D geometry of molecules is central to applications in drug discovery, materials design, and molecular modeling. However, molecular geometry can change dramatically across environments (e.g., crystal lattice versus protein binding pocket). Existing generative approaches are typically environment-agnostic or require separate models for each environment, which limits generalization. We introduce *EVA-Flow*, a unified framework for environment-aware conformation generation. EVA-Flow combines a variational autoencoder with a flow matching decoder and incorporates environment information through a learned embedding. Across four environments including vacuum, protein-ligand docking, solvation, and crystal packing, EVA-Flow substantially improves generation accuracy through pretraining and unification. Analysis of shared molecules that appear in multiple environments further shows that EVA-Flow generates distinct, environment-specific conformations rather than memorizing a single geometry.

## 1 INTRODUCTION

Molecules do not exist in a single fixed shape. Consider the journey of a drug molecule: it may be synthesized and dissolved in a solvent, crystallized into a solid for storage, redissolved in the body, and finally bind to a target protein to take effect. At each stage, the surrounding environment, solvent, crystal lattice, or protein pocket, reshapes the 3D geometry of the molecule. As a result, the same compound can adopt very different conformations depending on context. These environment-dependent conformations are not mere structural details; they govern fundamental properties such as binding affinity (Weikl & Paul, 2014), solubility (Sobornova et al., 2024), and stability, which in turn determine the efficacy of drugs, the performance of materials (Cruz-Cabeza et al., 2020), and the reliability of molecular simulations.

Recent generative models have made strong progress in learning molecular conformations, but most remain environment-agnostic, assuming molecules exist in isolation (Xu et al., 2022; Hassan et al., 2024). Some methods incorporate environments, but only for specific tasks such as protein–ligand docking (Corso et al., 2022; 2024), and must be trained separately for each case. This results in fragmented solutions with limited transferability. These limitations highlight the need for a unified framework that generalizes across environments by leveraging the flexibility and scalability of modern generative approaches.

In this work, we propose *EVA-Flow*, a unified model that produces environment-aware molecular conformations. The model builds on a variational autoencoder (VAE) (Kingma et al., 2019), where the encoder embeds both the molecular graph and its environment into a latent representation. This latent variable captures molecule–environment interactions, similar to collective variables (Fiorin et al., 2013; Bhakat, 2022). The flow matching (FM) decoder then generate molecular conformations conditioned on the environment (Lipman et al., 2022)(Figure 1). Unlike prior approaches restricted to a single context, EVA-Flow generalizes across environments within a single model, learning conformational distributions in vacuum, protein-ligand docking, solvation, and crystal packing.

Our contributions are summarized as follows:

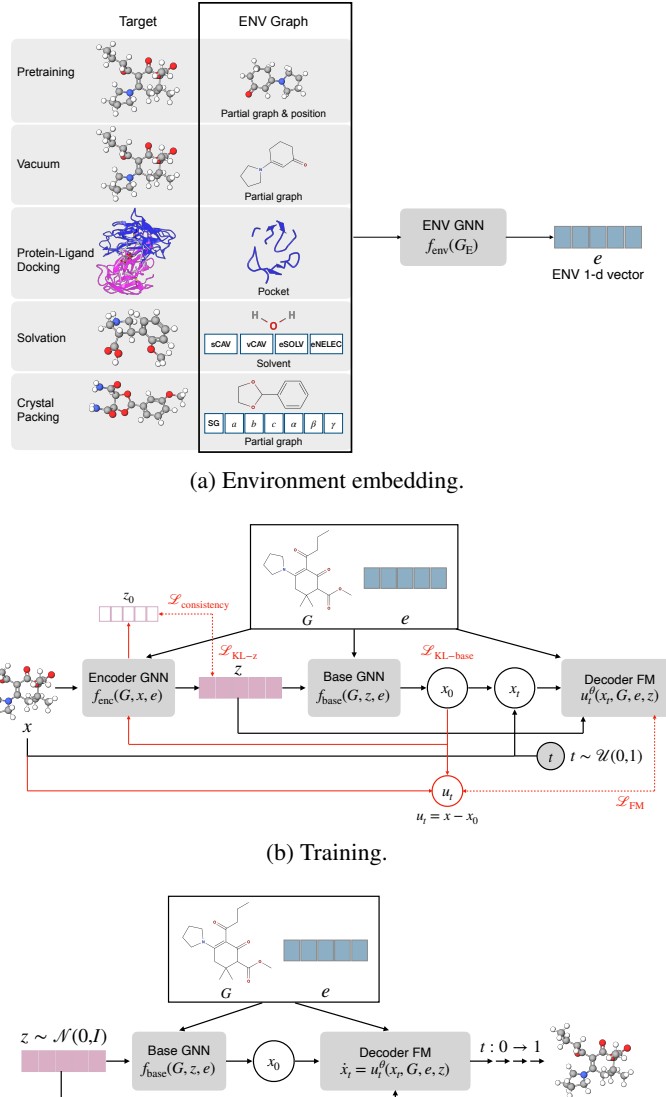

(a) Environment embedding.

(b) Training.

(c) Inference.

Figure 1: Architecture of EVA-Flow. (a) The raw environment graph $G_E$ is embedded by an environment network to produce a low-dimensional vector $e$. (b) At training time, the encoder $f_{enc}$ takes the molecular graph $G$, the environment vector $e$, and atomic positions $x$ to yield a latent variable $z$. A base network, conditioned on $(G, z, e)$, parameterizes a Gaussian base distribution $p_0(x \mid G, z, e)$ from which an initial conformation $x_0$ is sampled. A time is sampled $t \sim \mathcal{U}(0,1)$ and $x_t$ is calculated. The flow matching decoder learns a vector field $u_t^\theta(x_t, G, e, z)$. (c) At inference time, $z$ is sampled from $p(z) = \mathcal{N}(0, I)$, and then fed to the Base network. By solving the FM ODE from $x_{t=0} = x_0$, a conformer is generated.

- We introduce EVA-Flow, the first unified framework for environment-aware molecular conformation generation. EVA-Flow combines a VAE with a flow matching decoder.
- We design an encoder that integrates molecular and environmental information into a latent variable to capture molecule–environment interactions.
- We demonstrate that (i) pretraining and unification across environments improve accuracy over environment-specific models, and (ii) EVA-Flow generates distinct conformations for the same molecule in different environments, confirming that the model captures genuine environment dependence rather than memorizing a single geometry.

## 2 RELATED WORK

**Flow Matching**. FM trains a continuous normalizing flow by regressing the velocity field that transports a simple base distribution to the data along a chosen probability path, yielding a simulation-free objective and fast ODE sampling(Lipman et al., 2022). Rectified flow (RF) (Liu et al., 2022) and conditional FM (CFM) (Tong et al., 2023) extend FM to efficient straight-through paths and conditional generation. Equivariant (Hassan et al., 2024) and geometric variants (Chen & Lipman, 2023) adapt FM to 3D settings and enable geometric constraints and symmetries.

**VAE + FM**. Hybrid models combine VAE with flows in two ways. (1) Latent FM (better prior): FM is trained in the VAE latent space to transport a simple Gaussian to the aggregated posterior; sampling integrates the latent flow and then applies a standard feed-forward decoder, where FM improves the prior, not the decoder (Dao et al., 2023). (2) FM as decoder: the decoder itself is a conditional flow that maps a base to the data given the latent code, trained with FM/RF; this leads to few-step, deterministic sampling and better support for multi-modal outputs (Fischer et al., 2023; Sargent et al., 2025). Our work follows the latter design, instantiating an environment-aware FM decoder for conformer generation.

**Molecular conformation generation**. Generating 3D molecular structures has been extensively studied with both physics-based and learning-based approaches. Classical methods rely on force fields and molecular dynamics (MD) simulations, which are accurate but computationally expensive. Recent neural methods generate conformers with equivariant diffusion, flows, or autoregressive torsion models. Most papers consider vacuum (Xu et al., 2022; Hassan et al., 2024); a smaller body conditions on context such as protein pockets (protein-ligand docking) (Corso et al., 2024). However, these approaches typically target a single environment, requiring retraining or architecture changes to adapt across settings.

Key challenges in this space include: (i) the lack of unified models that generalize across diverse environments, and (ii) the difficulty of incorporating complex conditioning inputs such as structured environments into generative frameworks. EVA-Flow addresses these challenges by combining an environment-aware encoder with an SE(3)-equivariant FM decoder, enabling a single model to generate accurate conformers across different environments without requiring task-specific pipelines.

**Diffusion in latent space for molecules**. A recent work by Joshi et al. (2025) performs conformer generation by running a diffusion process in a learned latent space and decoding to coordinates. While this brings strong scalability and reuse of pretrained latents, it differs from our approach: we keep a VAE encoder but use FM as the decoder in data space, conditioned on the environment.

## 3 ENVIRONMENT-AWARE FLOW MATCHING

### 3.1 PROBLEM FORMULATION

Let $G = (V, E)$ denote a molecular graph with nodes representing atoms and edges representing bonds. Each atom is associated with a 3D position $x \in \mathbb{R}^{3N}$, where $N$ is the number of atoms. A molecule exists in an environment $E$, such as a protein pocket, solvent box, or crystalline lattice. Our goal is to learn the conditional distribution: $p(x|G, E)$, capturing how environments modulate molecular conformations. Unlike prior work that trains separate models for each environment, we seek a unified model that generalizes across all $E$.

### 3.2 MODEL ARCHITECTURE

Our framework, EVA-Flow, is a VAE with a FM decoder, consisting of multiple components (Figure 1).

1. **Environment Embedding Network**(Figure 1a). The environment is represented as a graph $G_{\mathrm{E}} = (V_{\mathrm{E}}, E_{\mathrm{E}})$, where nodes correspond to atoms with associated features and edges correspond to bonds or interactions. The construction of $G_{\mathrm{E}}$ is described in Section 4.1. The environment network, $f_{\mathrm{env}}$, consists of two graph convolutional network (GCN) layers (Zhang et al., 2019), followed by a two-layer MLP projection to the environment embedding vector $e \in \mathbb{R}^d$:

$$e = f_{\mathrm{env}}(G_{\mathrm{E}}) \tag{1}$$

This embedding provides global information that conditions both the encoder and the decoder.

2. **Encoder** (Figure 1b). The encoder is a GNN-based module that takes as input the molecular graph $G$, atomic positions $x$, and the environment embedding $e$. Input node features are formed by concatenating the atomic number, node attributes, and atomic positions. These features are processed by a stack of GCN layers, producing node embeddings of size $[N, \text{hidden}]$. Each node embedding is then concatenated with the environment embedding $e$, and projected through two linear layers to produce the mean and log-variance for node-level latent variables. This defines a variational posterior distribution

$$q_\phi(z \mid G, x, e) = \prod_{i=1}^{N} \mathcal{N}\Big(z_i \,\Big|\, \mu_{\phi,i}(G, x, e), \, \text{diag}\big(\sigma_{\phi,i}^2(G, x, e)\big)\Big), \tag{2}$$

where $\mu_{\phi,i}, \sigma_{\phi,i}^2 \in \mathbb{R}^d$ denote the mean and variance predicted for node $i$. Equivalently, stacking all nodes yields

$$q_\phi(z \mid G, x, e) = \mathcal{N}\big(\mu_\phi(G, x, e), \, \text{diag}(\sigma_\phi^2(G, x, e))\big), \tag{3}$$

with $\mu_\phi, \sigma_\phi^2 \in \mathbb{R}^{N \times d}$. The latent variables $z$ capture molecule–environment interactions.

3. **Base Distribution Network**. To initialize FM, we require a base distribution $p_0$. Instead of using a fixed isotropic Gaussian, we parameterize $p_0$ with a GNN, $f_{\text{base}}$, conditioned on the molecular graph $G$, the environment embedding $e$, and latent variables $z$. Specifically, input node features are constructed by concatenating the atomic number, node attributes, environment embedding, and latent variables. These features are processed by two GCN layers, followed by two independent MLP heads that predict the mean and log-variance of a Gaussian distribution:

$$\mu_{\text{base}}, \, \log \sigma_{\text{base}}^2 = f_{\text{base}}(G, z, e), \tag{4}$$

where $\mu_{\text{base}}, \sigma_{\text{base}}^2 \in \mathbb{R}^{N \times 3}$ correspond to the predicted mean and variance for the $N$ atoms in Cartesian space. This yields a factorized Gaussian base distribution over positions:

$$p_0(x \mid G, e, z) = \prod_{i=1}^{N} \mathcal{N}\Big(x_i \,\Big|\, \mu_{\text{base},i}(G, z, e), \, \text{diag}\big(\sigma_{\text{base},i}^2(G, z, e)\big)\Big). \tag{5}$$

4. **FM Decoder**. The decoder generates conformations by transporting samples from the base distribution $p_0$ to the target distribution. We adopt the ETFlow network (Hassan et al., 2024) to parameterize a time-dependent vector field $u_t^\theta$ that takes as input the current molecular structure $x_t$, the molecular graph $G$, the latent variables $z$, and the environment embedding $e$:

$$\dot{x}_t = u_t^\theta(x_t, G, z, e), \quad x_0 \sim p_0(x \mid G, z, e), \tag{6}$$

where $x_t \in \mathbb{R}^{3N}$ denotes the atomic coordinates at time $t \in [0, 1]$ that is linearly interpolated:

$$x_t = (1-t)x_0 + tx \tag{7}$$

5. **Inference**(Figure 1c). At inference time, given the environment graph $G_E$, we compute the embedding $e = f_{\text{env}}(G_E)$ and sample a latent $z \sim p(z) = \mathcal{N}(0, I)$. The base network $f_{\text{base}}(G, z, e)$ defines a Gaussian $p_0(x \mid G, z, e)$ from which we sample an initial conformation $x_0$. Starting at $x_{t=0} = x_0$, we solve the FM ODE (Equation (6)) to obtain the final conformer $\hat{x} = x_{t=1}$.

## 3.3 TRAINING OBJECTIVES

We jointly optimize the encoder, environment embedding network, base distribution network, and FM decoder using a composite objective. The total loss consists of four terms, each serving a distinct role (Figure 1b):

1. **Latent Prior Regularization**. This loss ensures the latent variable $z$ remains well-structured and consistent with a simple prior, preventing degenerate latents.

$$\mathcal{L}_{\text{KL}-z} = D_{\text{KL}}(q_\phi(z|x, G, e) \| \mathcal{N}(0, I)). \tag{8}$$

2. **Base Distribution Regularization**. This loss stabilizes the learned Gaussian base distribution and prevents drift and collapse of the base distribution.

$$\mathcal{L}_{\text{KL}-\text{base}} = D_{\text{KL}}(p_0(x|G, z, e) \| \mathcal{N}(0, I)), \tag{9}$$

.

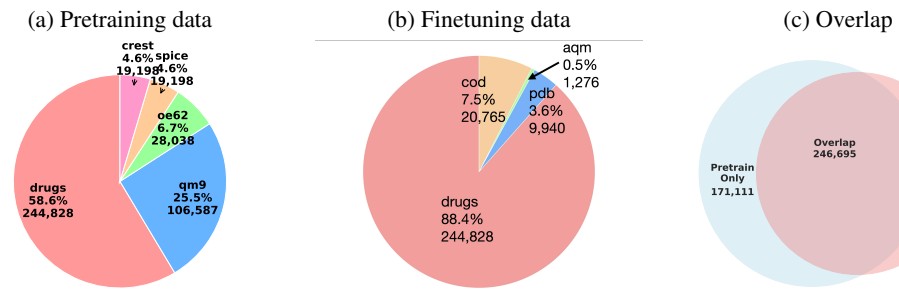

Figure 2: Dataset composition and overlap. (a) Pretraining data sources: GEOM-Drugs, GEOM-QM9, OE62, SPICE, and CREST-relaxed SPICE. Slice sizes are proportional to molecule counts. (b) Finetuning data composition across environments: vacuum (GEOM-Drugs), docking (PDB-Bind), packing (COD), and solvation (AQM). (c) Venn diagram showing the number of unique molecules in pretraining and finetuning datasets and their overlap.

3. **Latent Consistency Loss**. This loss encourages the latent representation $z$ to capture stable molecule-environment interactions. We compare the encoder's posterior means for the target conformation $x$ and the initial conformation $x_0 \sim p_0(\cdot|G, z, e)$:

$$\mathcal{L}_{\text{consistency}} = ||\mu_\phi(x_0, G, e) - \mu_\phi(x, G, e)||_2^2. \tag{10}$$

4. **FM Loss**. This loss trains the decoder to transport samples from the base distribution to the data distribution through continuous-time dynamics.

$$\mathcal{L}_{\text{FM}} = \mathbb{E}_{t, x_t}[||u_t^\theta(x_t, G, z, e) - u_t(x_0, x)||^2], \tag{11}$$

where $u_t$ is the analytically defined target velocity field:

$$u_t = x - x_0 \tag{12}$$

All networks are trained jointly at each iteration. We find that weighting the losses is not necessary since they are training orthogonal or complementary components, so our final loss is a sum.

$$\mathcal{L} = \mathcal{L}_{\text{KL-z}} + \mathcal{L}_{\text{KL-base}} + \mathcal{L}_{\text{consistency}} + \mathcal{L}_{\text{FM}}. \tag{13}$$

## 4 EXPERIMENTS

### 4.1 DATASETS

**Pretraining**. For large-scale pretraining, we combine datasets containing conformations of small organic and drug-like molecules. This includes relaxed conformations from GEOM-QM9 and GEOM-Drugs (Axelrod & Gomez-Bombarelli, 2022), as well as higher-energy conformations from the SPICE PubChem subset (Eastman et al., 2023). We further performed relaxation on SPICE dataset using `CREST` (Pracht et al., 2024) to obtain low-energy conformations. In addition, we include the OE62 dataset (Stuke et al., 2020), which consists of crystal-forming molecules relaxed in vacuum. Altogether, the pretraining corpus contains ∼418K molecules and 9.8M conformations (Figure 2a).

We construct the environment graph ($G_{\text{E}}$) during pretraining. For each molecule, we apply the RDKit (Landrum & contributors, 2025) `MurckoScaffold` package (RDK, 2025) to extract the molecule's Bemis–Murcko scaffold (Bemis & Murcko, 1996) by removing substituents and retaining only the core ring systems and the linkers connecting them. The partial graph, together with the atomic positions, is passed to the environment network (Figure 2a).

**Finetuning**. We finetune and evaluate our model in four environments:

1. **Vacuum**. In vacuum, the molecules are isolated and do not interact with external environment. We use GEOM-Drugs dataset. To build the environment graph, we extract the partial graph similar to the pretraining stage, but the atomic positions are all set to zero (Figure 1a, Vacuum).

2. **Protein–Ligand Docking**. In this task, molecules (ligands) adapt to protein binding pockets. We use PDBBind-v2020 dataset (Liu et al., 2015). The environment graph is constructed from heavy atoms in the pocket. We add edges between atoms in the same protein residue. The atomic positions of the binding pocket are included(Figure 1a, Protein-Ligand Docking).

3. **Solvation**. Dissolved molecules interact with the solvent molecules. To learn to predict the conformations in solution, we use the Aquamarine (AQM) dataset (Medrano Sandonas et al., 2024), which contains conformations relaxed in implicit water. To construct the environment graph, we use a water molecule and add solvation descriptors to the node features such as cavity surface area (sCAV), cavity volume (vCAV), free energy in electrolyte (eSOLV), and non-electrostatic free energy (eNELEC)(Figure 1a, Solvation). These descriptors provides information on the interaction between molecules and solvent.

4. **Crystal Packing**. Molecules interact with neighboring molecules in crystals. We use the Crystallography Open Database (COD) (Gražulis et al., 2009; 2012), and filter to the small organic molecular crystals. We convert the fractional coordinates from the CIF files to Cartesian positions, and recover the bonds with `OpenBabel` (O'Boyle et al., 2008; 2011). For the environment graph, we extract the partial graph and set atomic positions all to zero. We also add to the node feature the space group number (sg) and lattice parameters $(a, b, c, \alpha, \beta, \gamma)$ (Figure 1a, Packing).

Further details about data processing and the distributions of number of atoms and the number of rotatable bonds can be found in Appendices A and B.

### 4.2 EXPERIMENTAL SETUP

We compare four training strategies.

1. **Pretraining + Individual Finetuning**. We first pretrain the model on the pretraining datasets, then finetune it on a single environment. In the pretraining, the model is exposed to massive molecules and conformations to learn valid conformations. In the individual finetuning, the model learns a specific environment.

2. **Pretraining + Unified Finetuning**. We first perform training on the pretraining datasets (Section 4.1), but then finetune on all environments of the finetuning datasets. This setup examines if model learns to adapt to different environments.

3. **Unified Finetuning**. we skip the pretraining and only train the model across all environments.

4. **Individual Finetuning**. We train separate models for each environment, without pretraining. This setup serves as a baseline to study the effects of unification and pretraining.

To study the effects of scaling, we also explore three models of varying number of parameters: small (S, 9.4M), medium (M, 29.7M), and large (L, 68.3M).

**Evaluation metrics**. We evaluate generated conformers against ground-truth conformers using distance-based RMSD. We report the Recall and Precision Coverage (COV) and average minium RMSD (AMR) (Appendix D). For crystal packing, we also align the generated conformation with the ground-truth one and compute a symmetry-aware RMSD (sRMSD). This value reflects the minimal atomic displacement after accounting for lattice symmetry and periodic images.

### 4.3 RESULTS

We evaluate conformation generation in four environments under four training setups. The results are summarized in Table 1 (five-layer environment) and Table 4 (two-layer environment).

**Pretraining Improves Generalization**. Across all environments and for large-sized model, Pretrain+Individual and Pretrain+Unified setups consistently outperform both Unified (no pretraining) and Individual setups (Table 1, Figures 3a and 3c). The effects are less pronounced for Vacuum environment since the dataset is large and dominates the pretraining datasets. For the small and medium models (Tables 5 and 6, Figures 8 and 9), however, the Individual setup can beat the Pretrained one in Vacuum and Packing, perhaps due to limited capacity of the model.

**Unified vs. Individual Models**. The Unified setup generally performs better than the Individual one, but the Pretrain+Individual setup is better than the Pretrain+Unified one (Table 1). The results

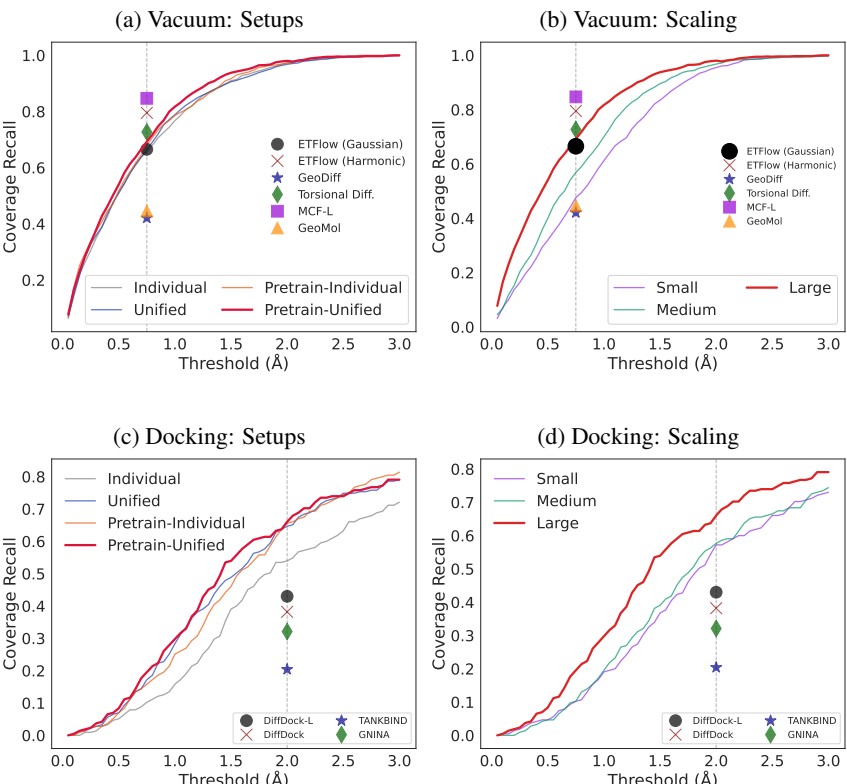

Figure 3: Recall COV for Vacuum and Docking. (a) Vacuum: comparison of four training setups: Individual (trained on GEOM-Drugs dataset only), Unified (joint training on Docking, Vacuum, Packing, and Solvation), Pretrain+Unified (unified initialized with pretraining), and Pretrain+Individual (pretraining and then on GEOM-Drugs dataset only). (b) Vacuum: effect of model sizes under Pretrain+Unified setup (S/M/L). The reference data points in (a) and (b) denote the results from Hassan et al. (2024); Xu et al. (2022); Jing et al. (2022); Wang et al. (2024); Ganea et al. (2021). (c) Docking: comparison of four training setups: Individual (trained on PDBBind dataset only), Unified, Pretrain+Unified, and Pretrain+Individual. (d) Docking: effect of model sizes under Pretrain+Unified setup (S/M/L). The reference data points in (c) and (d) denote the results from Corso et al. (2022; 2024); Lu et al. (2022); McNutt et al. (2021).

suggest that pretraining and unification can similarly help the model learn valid conformations, but unification can lead to competitions among different environments. A more powerful environment network and a larger environment embedding size may address this.

**Scaling**. Increasing model size yields clear improvements (Figures 3b, 3d, 8 and 9).

## 4.4 ANALYSIS

We quantify cross–environment shared molecules to probe whether the model truly conditions on the specified environment rather than memorizing a single geometry. The overlaps are small but non-trivial (e.g., 12 molecules appear in both Solvation and Docking, Figure 4a), which enables us to compare reference and generated conformers for the same molecule across environments. Some example visualizations can be found in Tables 2 and 7 from the Pretrain+Unified setup.

**Environment awareness when references differ**. For Solvation–Docking (Figure 4c), the two reference conformers for the same molecule differ substantially (2.33Å). In contrast, each generated conformer is close to its own environment's reference (Solvation: 0.64Å; Docking: 0.95Å), while cross–environment comparisons remain large (> 2.3Å). Similar results are observed for Solvation-

Table 1: Conformation generation across four environments using a five-layer environment network: Vacuum, Docking, Solvation, and Packing. We report mean Recall and Precision COV and AMR at environment-specific RMSD thresholds: 0.5Å for Solvation, 0.75Å for Vacuum and Packing, and 2.0Å for Docking. For Packing, we also report symmetry-aware RMSD (sRMSD). Best results are **bolded**. EVA-Flow results are compared to literature results for Vacuum and Docking environments.

| Environment | Setup | Recall | | Precision | | Crystal |
|---|---|---|---|---|---|---|
| | | COV↑ | AMR↓ | COV↑ | AMR↓ | sRMSD↓ |
| Vacuum | GeoDiff | 0.4210 | 0.835 | 0.2490 | 1.136 | – |
| | GeoMol | 0.4460 | 0.875 | 0.4300 | 0.928 | – |
| | ETFlow (Gaussian) | 0.6653 | 0.640 | 0.4441 | 0.903 | – |
| | Torsional Diff. | 0.7270 | 0.582 | 0.5520 | 0.778 | – |
| | ETFlow (Harmonic) | 0.7953 | 0.452 | **0.7438** | **0.541** | – |
| | MCF-L | **0.847** | **0.390** | 0.668 | 0.618 | – |
| | Pretrain+Unified | 0.6927 | 0.621 | 0.3498 | 1.228 | – |
| | Pretrain+Individual | 0.6754 | 0.622 | 0.3327 | 1.237 | – |
| | Unified | 0.6658 | 0.635 | 0.3340 | 1.232 | – |
| | Individual | 0.6563 | 0.656 | 0.3246 | 1.250 | – |
| Docking | TankBind | 0.204 | – | – | – | – |
| | GNINA | 0.321 | – | – | – | – |
| | DiffDock | 0.382 | – | – | – | – |
| | DiffDock-L | 0.430 | – | – | – | – |
| | Pretrain+Unified | **0.6605** | **1.933** | **0.4663** | **2.538** | – |
| | Pretrain+Individual | 0.6512 | 2.009 | 0.4605 | 2.597 | – |
| | Unified | 0.6465 | 1.975 | 0.4500 | 2.567 | – |
| | Individual | 0.5385 | 2.302 | 0.3182 | 2.918 | – |
| Solvation | Pretrain+Unified | **0.9427** | **0.175** | 0.6927 | 0.466 | – |
| | Pretrain+Individual | 0.9236 | 0.186 | **0.7245** | **0.409** | – |
| | Unified | 0.9172 | 0.182 | 0.6624 | 0.489 | – |
| | Individual | 0.4968 | 0.555 | 0.2006 | 0.921 | – |
| Packing | Pretrain+Unified | **0.5820** | **0.724** | **0.3602** | **1.149** | **0.061** |
| | Pretrain+Individual | 0.5280 | 0.806 | 0.3090 | 1.214 | 0.062 |
| | Unified | 0.5730 | 0.744 | 0.3455 | 1.179 | 0.064 |
| | Individual | 0.3820 | 0.998 | 0.2150 | 1.419 | 0.074 |

Table 2: Visualization of reference and generation conformations in Pretrain+Unified setup for shared molecules in Solvation-Packing and Solvation-Docking. The conformers are aligned to the reference of Solvation.

| Solvation-Packing | | | | Solvation-Docking | | | |
|---|---|---|---|---|---|---|---|
| ref_solv | gen_solv | ref_pack | gen_pack | ref_solv | gen_solv | ref_dock | gen_dock |

Packing (Figure 4b). Together, these trends indicate that the model adapts the conformation to the requested environment rather than collapsing to a single geometry.

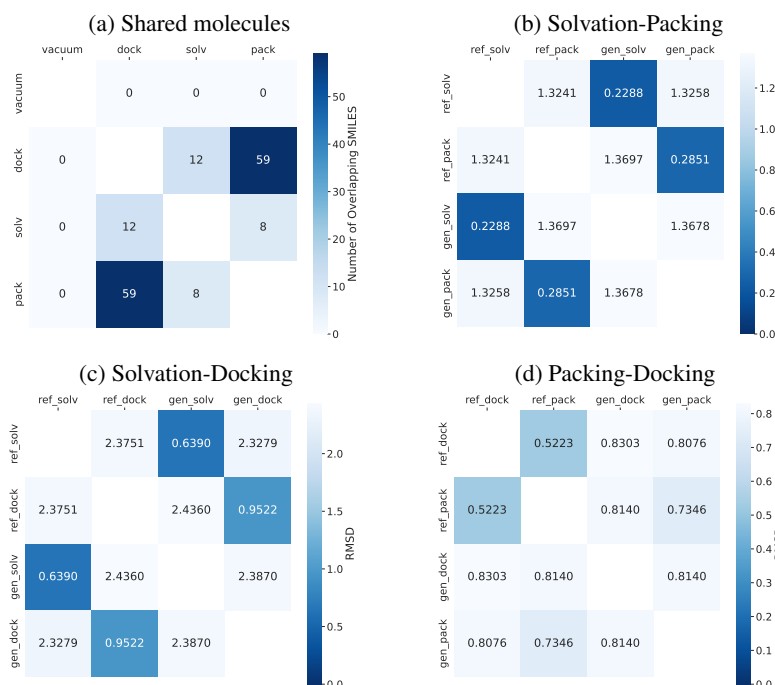

Figure 4: Shared-molecule analysis across environments. (a) Pairwise counts of molecules shared between environments. Darker cells indicate more molecules. (b-d) Pairwise heavy-atom RMSD (Å) between reference and generated conformers shared across: (b) Solvation-Packing (c) Solvation-Docking, and (d) Packing-Docking. In each heatmap we report the mean RMSD after alignment for: ref_A ↔ ref_B, gen_A ↔ ref_A, gen_A ↔ ref_B, gen_B ↔ ref_A, gen_B ↔ ref_B, and gen_A ↔ gen_B (A/B = the two tasks in the panel). Darker cells indicate lower RMSD (better agreement).

**When environments agree, generations agree**. For Packing–Docking (Figure 4d), the two references are already similar ($0.52$Å), and the generated conformers across the two environments are also close ($0.81$Å), consistent with the underlying agreement between environments.

These patterns provide direct evidence that the model is environment-aware rather than memorizing a single canonical geometry.

## 4.5 ABLATION STUDIES

We ablate key components of our model to quantify their contributions to conformer generation across the four environments, using the Unified setup and small model. The full system (Full) includes model components and training objective as defined in Sections 3.2 and 3.3. We consider three ablations: 'NoConsist' removing the latent consistency loss term; 'NoBase' replacing the base distribution network by isotropic Gaussian noise for $x_0$; 'NoFM' removing the FM decoder and using the base distribution network as the decoder with a regression loss.

Figure 5 reports Recall COV of these ablations for all environments. The full system achieves the highest Recall in every environment. The ablation of the FM decoder consistently underperforms across all environments, because regression tends to learn the conditional mean of a multi-modal conformer distribution, leading to blurred geometry and lower coverage. The ablation of the base distribution network (NoBase) yields a large degradation, indicating that a learned, environment-aware base distribution for $x_0$ is essential for capturing environment-molecule interactions. The ablation of latent consistency loss (NoConsist) produces a smaller but repeatable drop in Recall COV, suggesting that the latent consistency term stabilizes training and provides regularization.

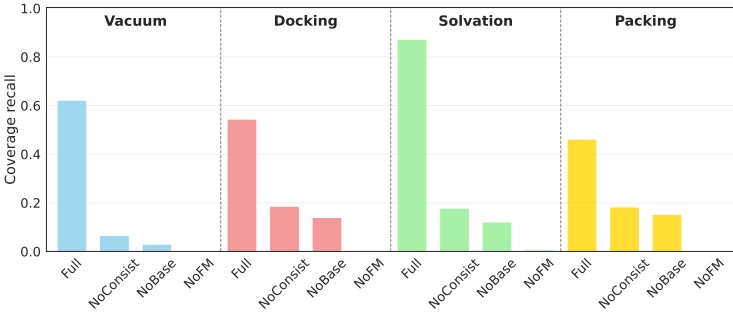

Figure 5: Recall COV for all environments (Vacuum, Docking, Solvation, Packing) using the Unified setup for small model. 'Full' refers to full model; 'NoConsist' is removing latent consistency loss; 'NoBase' is for replacing the learned base distribution network for initial positions $x_0$ with isotropic Gaussian noise; 'NoFM' is removing the FM decoder and using a regression decoder.

## 5 CONCLUSION

We introduced EVA-Flow, a unified framework for environment-aware molecular conformation generation that couples a VAE with an environment-conditioned flow-matching decoder and a learned base distribution. Experiments show that EVA-Flow generates more accurate conformers across vacuum, protein–ligand docking, solvation, and crystal packing. Analyses of molecules shared across environments confirm that EVA-Flow produces distinct, environment-specific conformations rather than collapsing to a single geometry.

While EVA-Flow demonstrates the feasibility of unified, environment-aware conformation generation, several limitations remain and point to valuable future directions. First, we focused on a specific architecture: a single FM decoder (ETFlow) with GCN-based networks for the base, encoder, and environment modules. More expressive architectures may further improve generalization and accuracy. Second, our base distribution assumes a simple Gaussian prior; extending it to more structured priors such as harmonic prior could better capture the geometry of conformers. Third, while EVA-Flow is designed to scale, our current models were of modest size. Increasing the capacity of the environment encoder and latent embeddings could further enhance the model's ability to capture complex molecule-environment interactions. Finally, our evaluation was limited to four environments with high-quality datasets. Extending EVA-Flow to more diverse and challenging contexts, such as surface adsorption on catalysts or solid–liquid interfaces, will require new benchmarks with carefully curated data and poses an exciting direction for future work.

## 6 REPRODUCIBILITY STATEMENT

All datasets used in this work are either open source or processed with open-source software. Details of the data processing steps are provided in the Appendix. The models we use are also open source. We describe the model architecture in the main text and list all hyperparameters in the Appendix.

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

## A   DATA PREPARATION

**GEOM QM9 and Drugs**    For GEOM QM9 and Drugs datasets, we used the data processing script provided by the ETFlow work. We used the same splitting of train, validation, and test. For each molecule, we kept the top 30 conformers with the highest Boltzmann weights. Each molecule was saved as a single PyTorch Geometric object with attributes atomic numbers, atomic positions, and SMILES string. The resulting QM9 training set includes 106,587 molecules with 794,960 conformations, QM9 validation set includes 13,323 molecules with 99,945 conformations. The test set has 1,000 molecules. The resulting Drugs training set has 244,828 molecules with 5,770,377 conformations and the validation set has 30,330 molecules with 715,592 conformations. The test set has 1,000 molecules.

**SPICE and CREST-relaxed SPICE**    We used the PubChem subset of the SPICE dataset (Eastman et al., 2023). This subset includes small, drug-like molecules comprised of between 18 and 50 atoms with elements Br, C, Cl, F, H, I, N, O, P, and S.

Since the conformers in SPICE dataset are not fully relaxed, we build a relaxed set for the molecules in SPICE dataset. For each molecule, we first used RDKit to generate 50 conformers with `EmbedMultipleConfs` using parameters `pruneRmsThresh=0.01`, `maxAttempts=5`, `useRandomCoords=False` corresponding to a pruning threshold of similar conformers of 0.01Å, a maximum of five embedding attempts per conformer, coordinate initialization from the eigenvalues of the distance matrix, and a random seed. If no conformers were successfully generated then `numConfs` was increased to 500. Afterwards, the conformers were optimized in the RDKit default MMFF force field. We deduplicated the conformers using RDKit's `GetBestRMS`, unless the calculation took longer than 48 hours then we switched to RDKit's `AlignMol`. After removing the duplicate conformers that exhibited an RMSD $< 0.1$Å, the ten conformers with the lowest energy were further optimized with an approximate energy model known as *extended tight binding* (xTB) (Grimme et al., 2017; Friede et al., 2024). The conformer with the lowest xTB energy was used as input to the CREST simulation. We used the default hyperparameters from CREST version 3.0.2 including a 6 kcal/mol cutoff on final conformers. Any SMILES strings containing slash or backslash indicate a cis/trans stereochemistry were skipped since their SMILES string is not preserved during the above procedure.

For both the original SPICE dataset and the CREST-relaxed dataset, we prepare the data in the PyTorch Geometric Data objects. For each molecule, we built an RDKit molecule object from SMILES string. However, the order of atoms extracted from the RDKit molecule object can be different from the atomic positions in the datasets. To fix this order issue, we used RDKit's GetSubstructMatch function to build a mapping. We make sure the atomic positions align with the atomic numbers and the RDKit molecule built from the SMILES. These three components make the PyTorch Geometric Data object.

We did a random splitting of the datasets at 0.8:0.1:0.1 to train, validation, and test. The resulting train set has 19,198 molecules with 959,802 and 2,203,392 conformers for SPICE and CREST, respectively. The validation set has 2508 molecules with 125,370 and 278,673 conformers for SPICE and CREST. The test set has 2,941 molecules.

**OE62**    We prepare the data in the PyTorch Geometric Data objects. For each molecule, we read the SMILES string and PBE-relaxed coordinates from the dataset. We build an RDKit molecule object from the SMILES string. Similar to SPICE dataset, the order of atoms extracted from the RDKit molecule object can be different from the coordinates. We used RDKit's GetSubstructMatch function to build a mapping to make sure the atomic positions align with the order of the RDKit molecule. The PyTorch Geometric Data object consists of SMILES string, coordinates, and atomic numbers.

We did a random splitting of the datasets at 0.8:0.1:0.1 to train, validation, and test. The resulting train set has 28,038 molecules and the validation set has 4,297 molecules, and the test set has 3,933 molecules. Each molecule has one conformation.

**PDBBind**    We use the PDBBind-v2020 dataset. For each protein-ligand pair, we read the ligand sdf file and use RDKit's Chem.SDMolSupplier to build an RDKit molecule object. We then add

hydrogen to the molecule and write a canonical SMILES. Then we use RDKit to make a molecule from the SMILES string. We notice that these two molecule objects, one from sdf file and the other from SMILES string, generate different orders of atoms. Therefore, we use the trick of RDKit's GetSubstructMatch to align the two molecule objects and find the order of coordinates that match the order of atoms in the RDKit molecule built from SMILES string. The atomic numbers are extracted from the RDKit molecule built from SMILES string.

We take the protein's binding pocket as the environment graph. We used PDBParser to read pocket pdb file. We only include heavy atoms in the graph. We add an edge between two heavy atoms if they are within one residue. We add node features using ETFlow's approach by setting chirality, degree, formal charge, implicit valence, number of hydrogen, hybridization, and number of radical electrons all to "misc", and no aromaticity or rings.

The final PyTorch Geometric Data object includes ligand's SMILES string, list of atomic numbers, atomic positions, and protein pocket's list of atomic numbers, node features, edge indices, and atomic positions.

We inherit the splitting of the PDBBind dataset. After skipping examples that the ligands fail to generate a valid RDkit molecule, the resulting train set has 9,940 examples and the validation set has 614 examples, and the test set has 218 examples. Each example has one conformation for the ligand.

**Aquamarine (AQM)**   We read SMILES string from the dataset and build an RDKit molecule object from the SMILES string. We add hydrogen to the molecule. We then check if the given order of atoms agrees with the order extracted from the RDKit molecule. If not, we build a second RDKit molecule from the given coordinates and use the RDKit's GetSubstructMatch to map the coordinates to the order of the SMILES-generated RDKit molecule.

In addition, to construct the raw environment graph, we use water molecule's graph since the dataset is collected in the presence of implicit water. We use ETFlow's approach to generate node feature and edge indices. We also include four descriptors that summarizes the interaction between the solute molecule and water solvent:

- sCAV: surface area of cavity;
- vCAV: volume of cavity;
- eSOLV: free energy in electrolyte;
- eNELEC: non-electrostatic free energy.

We normalize each descriptor by subtracting the mean of that descriptor across the dataset and then divided by the standard deviation. These descriptors are added to the node features for each node.

After filtering examples that failed generating valid RDKit molecule objects, the final PyTorch Geometric Data object include the solute molecule's SMILES string, atomic numbers, and coordinates, as well as water molecule's atomic numbers, node features, and edge indices. The coordinates of the water atoms are all set to zero.

We did a random splitting of the datasets at 0.8:0.1:0.1 to train, validation, and test. The resulting train set has 1,276 molecules and the validation set has 175 molecules, and the test set has 157 molecules. Each molecule has one conformation.

**Crystallography Open Database (COD)**   (Gražulis et al., 2009). According to their website, the COD is "An open-access collection of crystal structures of organic, inorganic, metal-organic compounds and minerals, excluding biopolymers." We utilize the organic crystals in this dataset for our so-called *packing* fine-tuning experiment. One may think of a molecular crystal as a conformer of an organic molecule that has been placed into a periodic lattice, where copies of this conformer are placed at regular distances extending infinitely over all of space. The periodicity implies that there a repeating smallest tile, or so-called asymmetric unit. This is necessary so that we can define a notion of local structure in our environment embedding. We now describe the nature of our subset.

We considered all of the structures in COD, then we applied the following filters:

- The structure must contain organic molecules.

- The structure must not be marked as disordered. "Disorder is a violation of crystal symmetry, where an atom is distributed over several positions or shared by several atoms, resulting in an average structure." (Groom et al., 2016).

- The structure must not be marked as missing Hydrogen.

- The structure is allowed to contain metals.

- The structure must have SMILES (Weininger, 1988).

- There must be only one fragments in the SMILES, defined by having no . in the string.

- The structure must pass `clean_st` (Cao et al., 2024).

- The structure must have only one fragment according to `extractStructure` (Cao et al., 2024).

- The structure must contain only one formula unit (molecule) in the asymmetric unit (a property typically called $Z' = 1$) according to `renumber_molecules_to_match`, `group_with_comparison`, and `get_rmsd_allowing_refl` (Cao et al., 2024).

We determined the topology using `renumber_molecules_to_match` rather than relying on the SMILES strings provided by the COS since they are often unreliable. Furthermore, extracted the space group, Wyckoff position, and atomic coordinates of each formula unit (molecule). This enabled us to construct its environment for embedding. The result is a set of 20,765 molecular crystals with symmetry information and a single formula unit (molecule) in the asymmetric unit.

We then process the examples into PyTorch Geometric Data objects. We read the sdf files from the dataset, and build RDKit molecule object using Chem.MolFromMolBlock. A SMILES string is then generated from Chem.MolToSmiles, which is then used to build a second RDKit molecule object. We check if the order of atoms in the two molecule objects are the same. If not, we use the GetSubstructMatch trick to map them and obtain the atomic positions in the same order of atoms as the molecule object from SMILES string.

We also obtain lattice parameters including the unit cell's lengths $(a, b, c)$ and angles $(\alpha, \beta, \gamma)$ and space group number from the dataset.

The molecule's SMILES string, list of atomic numbers, coordinates, and the lattice parameters are included in the PyTorch Geometric Data object.

After filtering examples that fail to generate valid RDkit molecule object, we did a random splitting of the dataset at 0.8:0.1:0.1 to train, validation, and test. The resulting train set has 20,765 molecules and the validation set has 3,239 molecules, and the test set has 3,235 molecules. Each molecule has one conformation.

## B    DISTRIBUTION OF NUMBER OF ATOMS AND NUMBER OF ROTATABLE BONDS

To characterize the intrinsic difficulty of conformer generation across datasets, we report histograms of molecular size (number of atoms, $N_{\text{atoms}}$) and flexibility (number of rotatable bonds, $N_{\text{rot}}$) for each dataset (Figures 6 and 7). We compute $N_{\text{rot}}$ using `rdMolDescriptors.CalcNumRotatableBonds` from RDKit with its default definition (non-ring single bonds between non-terminal heavy atoms).

Across the pretraining sets (GEOM Drugs, GEOM QM9, SPICE, CREST-relaxed SPICE, and OE62), $N_{\text{atoms}}$ concentrates in the mid-size regime with means in the $\sim 17$–$44$ range, while $N_{\text{rot}}$ is typically modest (most molecules have fewer than 15 rotatable bonds).

For the finetuning sets (GEOM Drugs, PDBBind, AQM, and COD), the atom-count distribution shifts larger with means in the $\sim 38 - 64$ range. In particular, the PDBBind dataset exhibits a long tail (very large molecules), indicating increased geometric and torsional complexity.

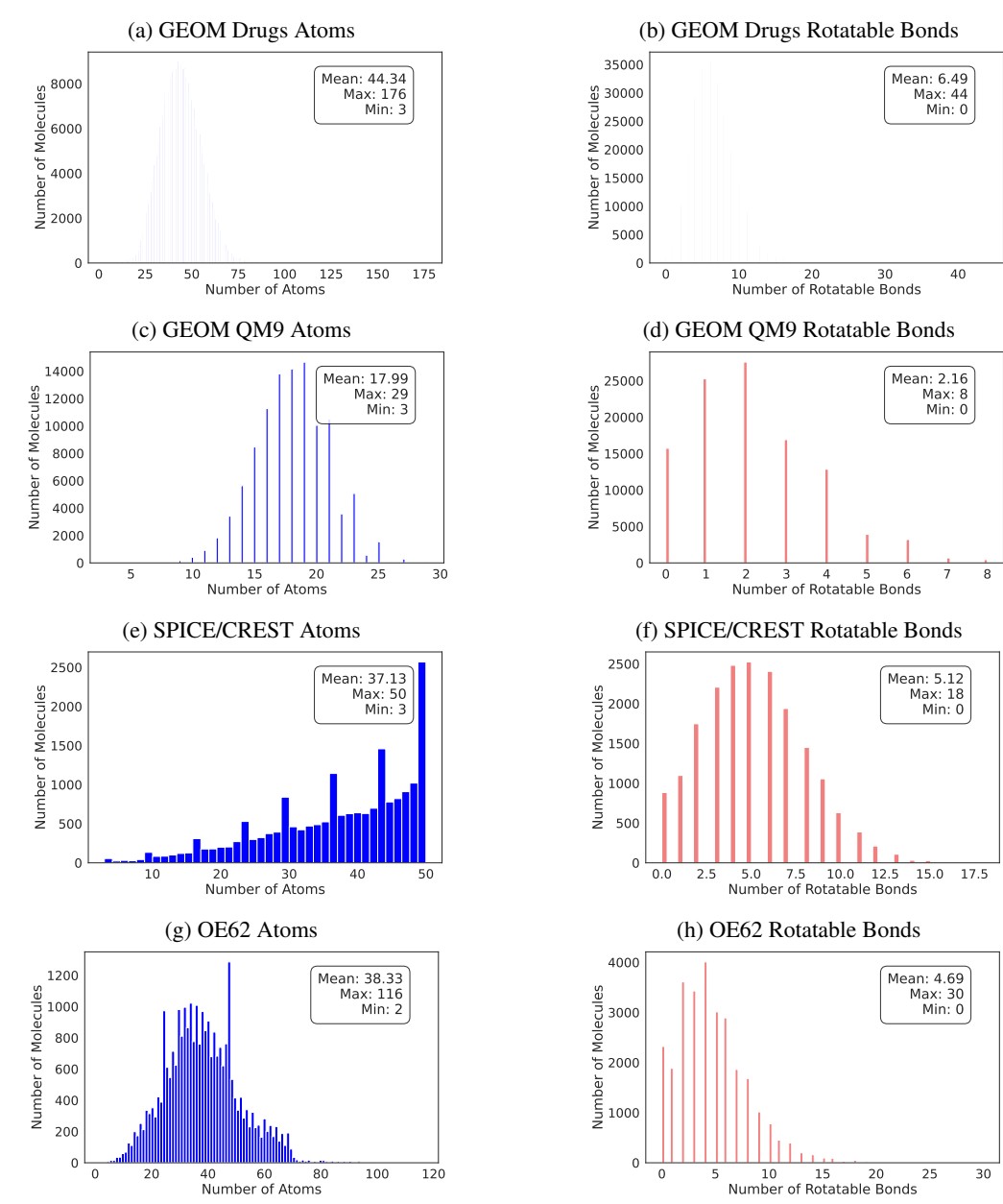

Figure 6: The histograms of the number of atoms and the number of rotatable bonds in the pretraining datasets, including GEOM Drugs, GEOM QM9, SPICE, CREST-relaxed SPICE, and OE62.

## C  TRAINING DETAILS AND HYPERPARAMTERS

For pretraining, we train EVA-Flow for a fixed 500 epochs. For each epoch, we randomly sample 50,000 examples from the pretraining datasets with a probability of 0.5 for GEOM-Drugs, 0.2 for OE62, 0.1 for GEOM-QM9, 0.1 for SPICE, and 0.1 for CREST. We use the epoch number as the random seed. For the learning rate, we use the Adam Optimizer with a cosine annealing learning rate which goes from a maximum of $8 \times 10^{-4}$ to a minimum $10^{-7}$ over 500 epochs with a weight decay of $10^{-10}$. We use 8 A100 GPUs.

For unified finetuning, we train EVA-Flow for a fixed 500 epochs. For each epoch, we randomly sample 50,000 examples from the pretraining datasets with a probability of 0.6 for GEOM-Drugs (Vacuum), 0.2 for COD (Packing), 0.15 for PDBBind (Docking), and 0.05 for AQM (Solvation).

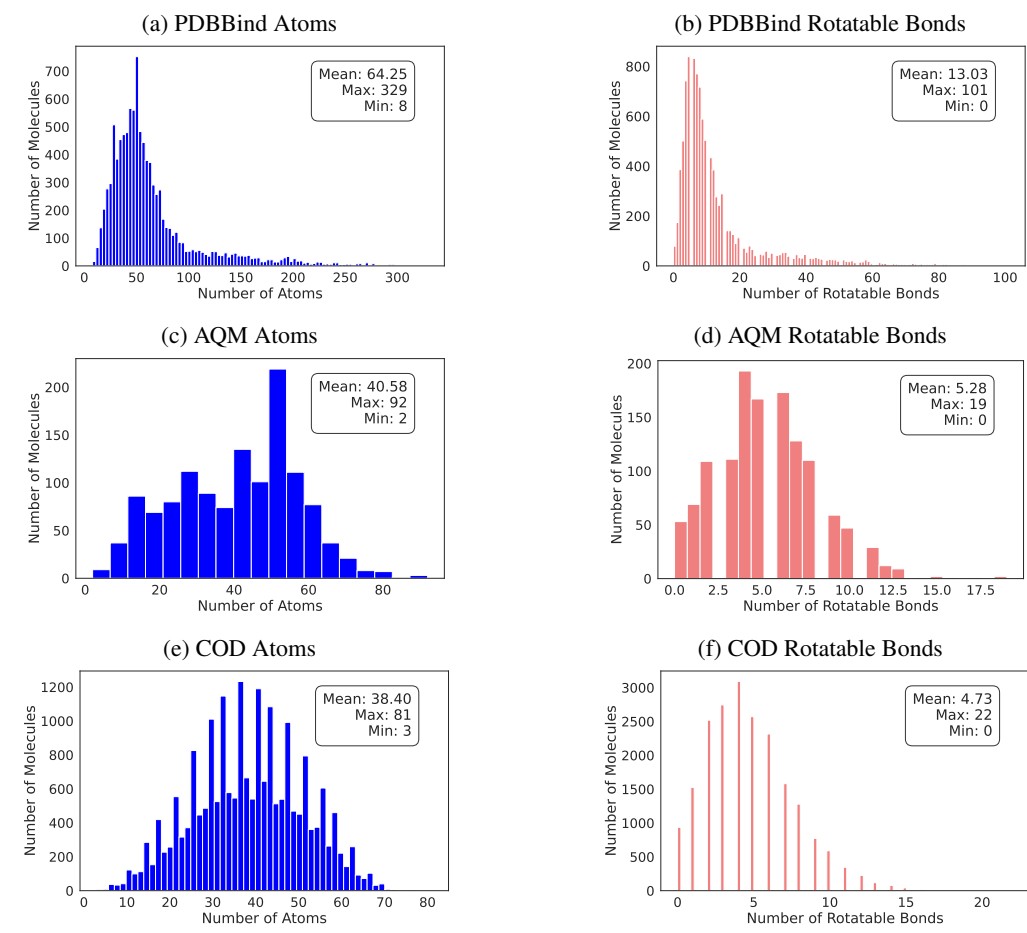

Figure 7: The histograms of the number of atoms and the number of rotatable bonds in the finetuning datasets, including PDBBind, AQM, and COD. The GEOM Drugs dataset is the same as the one in the pretraining datasets (Figure 6).

We use the epoch number as the random seed. For the learning rate, we use the Adam Optimizer with a cosine annealing learning rate which goes from a maximum of $8 \times 10^{-4}$ to a minimum $10^{-7}$ over 500 epochs with a weight decay of $10^{-10}$. We use 8 A100 GPUs.

For individual finetuning, we train EVA-Flow for a fixed 500 epochs on each full dataset. For the learning rate, we use the Adam Optimizer with a cosine annealing learning rate which goes from a maximum of $8 \times 10^{-4}$ to a minimum $10^{-7}$ over 500 epochs with a weight decay of $10^{-10}$. We use 8 A100 GPUs.

## D    EVALUATION METRIC DEFINITION

We compute average minimus RMSD (AMR) and Coverage (COV) Recall and Precision to assess the performance of molecular conformer generation following the approaches of (Ganea et al., 2021; Xu et al., 2022; Jing et al., 2022). Recall measures the extent to which the generated conformers capture the ground-truth conformers, while Precision indicates the proportion of generated conformers that match the ground-truth conformers. Using $C_g$ to denote the set of generated conformations, and $C_r$ the set of reference conformations, we calculate these metrics using the following equations:

$$\text{AMR-R}(C_g, C_r) = \frac{1}{|C_r|} \sum_{\mathbf{R} \in C_r} \min_{\hat{\mathbf{R}} \in C_g} \text{RMSD}(\mathbf{R}, \hat{\mathbf{R}}) \tag{14}$$

Table 3: Hyperparameters for EVA-Flow

| Hyper-parameter | Small | Medium | Large |
|---|---|---|---|
| bsz | 32 | 16 | 8 |
| num_layers | 20 | 30 | 40 |
| hidden_channels | 160 | 240 | 320 |
| num_heads | 8 | 12 | 16 |
| neighbor_embedding | True | True | True |
| cutoff_lower | 0.0 | 0.0 | 0.0 |
| cutoff_higher | 10.0 | 10.0 | 10.0 |
| node_attr_dim | 10 | 10 | 10 |
| edge_attr_dim | 1 | 1 | 1 |
| reduce_op | True | True | True |
| activation | SILU | SILU | SILU |
| attn_activation | SILU | SILU | SILU |
| latent_dim | 128 | 192 | 256 |
| env_dim | 128 | 192 | 256 |
| encoder_num_layer | 2 | 2 | 3 |
| base_num_layer | 2 | 2 | 2 |
| env_num_layer | 2 | 2 | 5 |
| FM decoder # param | 9.1M | 29.1M | 67.0M |
| encoder # param | 102K | 227K | 505K |
| env # param | 68.0K | 96.8K | 133K |
| base # param | 146K | 327K | 580K |
| total # param | 9.4M | 29.7M | 68.3M |

$$\text{COV-R}(C_g, C_r) = \frac{1}{|C_r|}|\{\mathbf{R} \in C_r | \text{RMSD}(\mathbf{R}, \hat{\mathbf{R}}) < \delta, \hat{\mathbf{R}} \in C_g\}| \tag{15}$$

$$\text{AMR-P}(C_r, C_g) = \frac{1}{|C_g|} \sum_{\hat{\mathbf{R}} \in C_g} \min_{\mathbf{R} \in C_r} \text{RMSD}(\hat{\mathbf{R}}, \mathbf{R}) \tag{16}$$

$$\text{COV-P}(C_r, C_g) = \frac{1}{|C_g|}|\{\hat{\mathbf{R}} \in C_g | \text{RMSD}(\hat{\mathbf{R}}, \mathbf{R}) < \delta, \mathbf{R} \in C_r\}| \tag{17}$$

A lower AMR score signifies improved accuracy, while a higher COV score reflects greater diversity in the generative model. The threshold $\delta$ is set to 0.5 Åfor Solvation, 0.75 Åfor Vacuum (Hassan et al., 2024) and Packing, and 2.0 Åfor Docking (Corso et al., 2022).

## E  MORE RESULTS

The performance of large models with a two-layer environment network, and small and medium models across four environments are presented in Table 4, Figures 8 and 9 and Tables 5 and 6.

## F  MOLECULE VISUALIZATION

The visualization of reference and generated conformers for molecules shared in the Packing-Docking environments.

## G  USE OF LLMS

We used large language models (LLMs) to check grammar, correct typos, and refine word usage.

Table 4: Conformation generation across four environments using a two-layer environment network: Vacuum, Docking, Solvation, and Packing. We report mean Recall and Precision COV and AMR at environment-specific RMSD thresholds: 0.5Å for Solvation, 0.75Å for Vacuum and Packing, and 2.0Å for Docking. For Packing, we also report symmetry-aware RMSD (sRMSD). Best results are **bolded**.

| Environment | Setup | Recall | | Precision | | Crystal |
|---|---|---|---|---|---|---|
| | | COV↑ | AMR↓ | COV↑ | AMR↓ | sRMSD↓ |
| Vacuum | Pretrain+Individual | 0.6794 | **0.617** | 0.3279 | 1.230 | – |
| | Pretrain+Unified | 0.6453 | 0.632 | 0.3234 | 1.236 | – |
| | Unified | 0.6224 | 0.690 | 0.3029 | 1.272 | – |
| | Individual | 0.6563 | 0.656 | 0.3246 | 1.250 | – |
| Docking | Pretrain+Individual | 0.6558 | 2.013 | 0.4500 | 2.617 | – |
| | Pretrain+Unified | 0.6233 | 2.033 | 0.4302 | 2.648 | – |
| | Unified | 0.5944 | 2.260 | 0.4266 | 2.882 | – |
| | Individual | 0.5385 | 2.302 | 0.3182 | 2.918 | – |
| Solvation | Pretrain+Individual | 0.9299 | 0.184 | **0.8790** | **0.265** | – |
| | Pretrain+Unified | 0.9172 | 0.190 | 0.7278 | 0.452 | – |
| | Unified | 0.8535 | 0.252 | 0.5175 | 0.617 | – |
| | Individual | 0.4968 | 0.555 | 0.2006 | 0.921 | – |
| Packing | Pretrain+Individual | 0.5280 | 0.789 | 0.3092 | 1.211 | **0.057** |
| | Pretrain+Unified | 0.4930 | 0.837 | 0.2785 | 1.263 | 0.069 |
| | Unified | 0.4400 | 0.885 | 0.2492 | 1.304 | 0.068 |
| | Individual | 0.3820 | 0.998 | 0.2150 | 1.419 | 0.074 |

Table 5: Conformation generation for Small models. We report Recall and Precision Coverage at environment-specific RMSD thresholds: 0.5Å for Solvation, 0.75Å for Vacuum and Packing, and 2.0Å for Docking. For Packing, we also report symmetry-aware RMSD (sRMSD).

| Environment | Setup | Recall | | Precision | | Crystal |
|---|---|---|---|---|---|---|
| | | Coverage↑ | AMR↓ | Coverage↑ | AMR↓ | sRMSD↓ |
| Vacuum | Pretrain+Unified | 0.4760 | 0.884 | 0.2082 | 1.464 | – |
| | Unified | 0.4349 | 0.941 | 0.1934 | 1.508 | – |
| | Individual | **0.5200** | **0.830** | **0.2355** | **1.405** | – |
| Docking | Pretrain+Unified | **0.5767** | 2.283 | **0.4035** | 2.893 | – |
| | Unified | 0.5442 | **2.277** | 0.3814 | 2.903 | – |
| | Individual | 0.5116 | 2.557 | 0.3186 | 3.198 | – |
| Solvation | Pretrain+Unified | 0.8662 | **0.236** | **0.5637** | **0.554** | – |
| | Unified | **0.8726** | 0.246 | **0.5637** | 0.562 | – |
| | Individual | 0.4968 | 0.558 | 0.2006 | 0.896 | – |
| Packing | Pretrain+Unified | **0.4830** | **0.861** | **0.2740** | **1.286** | **0.062** |
| | Unified | 0.4620 | 0.880 | 0.2645 | 1.299 | 0.064 |
| | Individual | 0.3580 | 1.022 | 0.2065 | 1.415 | 0.072 |

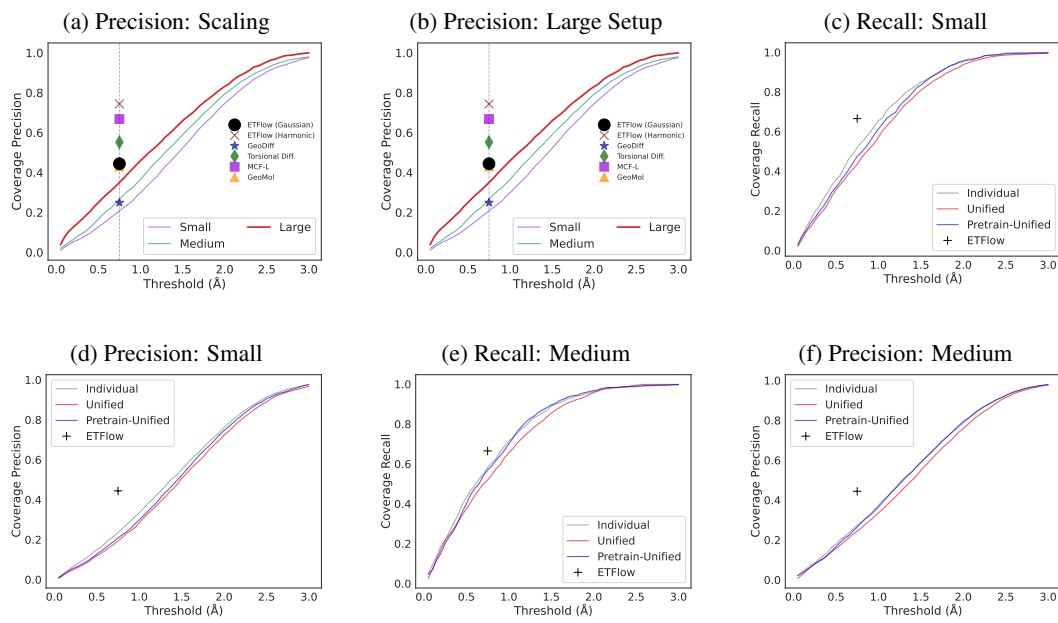

Figure 8: Recall and Precision Coverage for Vacuum. (a) Precision Coverage of three model sizes, small, medium and large. (b) Large model: comparison of Precision Coverage for three training setups: Individual (trained on PDBBind dataset only), Unified (joint training on Docking, Vacuum, Packing, and Solvation), and Pretrain+Unified (Unified initialized with pretraining). (c)(c) Small model: comparison of Recall and Precision Coverage for three training setups: Individual, Unified, and Pretrain+Unified. (c)(d) Medium model: comparison of Recall and Precision Coverage for three training setups: Individual, Unified, and Pretrain+Unified. The cross in (a) and (b) denotes the state-of-the-art result from Hassan et al. (2024).

Table 6: Conformation generation for Medium models. We report Recall and Precision Coverage at environment-specific RMSD thresholds: 0.5Å for Solvation, 0.75Å for Vacuum and Packing, and 2.0Å for Docking. For Packing, we also report symmetry-aware RMSD (sRMSD).

| Environment | Setup | Recall | | Precision | | Crystal |
|---|---|---|---|---|---|---|
| | | Coverage↑ | AMR↓ | Coverage↑ | AMR↓ | sRMSD↓ |
| | Pretrain+Unified | 0.5691 | 0.758 | 0.2650 | 1.337 | – |
| Vacuum | Unified | 0.5196 | 0.816 | 0.2447 | 1.402 | – |
| | Individual | **0.5822** | **0.748** | **0.2730** | **1.329** | – |
| | Pretrain+Unified | **0.5721** | **2.364** | **0.3744** | **2.987** | – |
| Docking | Unified | 0.5023 | 2.527 | 0.3372 | 3.161 | – |
| | Individual | 0.4791 | 2.566 | 0.3233 | 3.203 | – |
| | Pretrain+Unified | **0.8599** | **0.246** | **0.5780** | **0.575** | – |
| Solvation | Unified | 0.4841 | 0.539 | 0.2070 | 0.863 | – |
| | Individual | 0.5287 | 0.555 | 0.2245 | 0.892 | – |
| | Pretrain+Unified | 0.4110 | 0.929 | 0.2402 | 1.338 | **0.064** |
| Packing | Unified | 0.3250 | 1.078 | 0.1795 | 1.493 | 0.078 |
| | Individual | **0.4210** | **0.927** | **0.2412** | **1.332** | 0.068 |

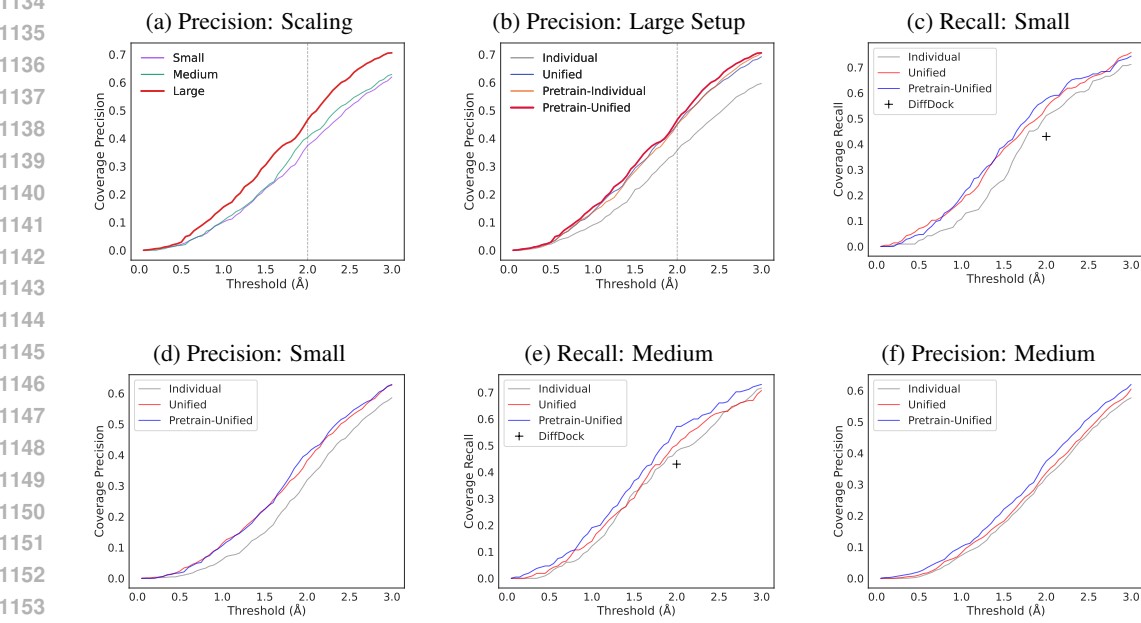

Figure 9: Recall and Precision Coverage for Docking. (a) Precision Coverage of three model sizes, small, medium and large. (b) Large model: comparison of three training setups: Individual (trained on PDBBind dataset only), Unified (joint training on Docking, Vacuum, Packing, and Solvation), and Pretrain+Unified (Unified initialized with pretraining). (c)(c) Small model: comparison of three training setups: Individual, Unified, and Pretrain+Unified. (c)(d) Medium model: comparison of three training setups: Individual, Unified, and Pretrain+Unified. The cross in (a) and (b) denotes the state-of-the-art result from Xu et al. (2022).

Table 7: Visualization of reference and generation conformations for the shared molecules in Packing-Docking. The molecules are aligned to the reference conformers of Docking environment.

| Packing-Docking | | | |
|---|---|---|---|
| ref_dock | gen_dock | ref_pack | gen_pack |

