# OpenReview forum: "EVA-Flow: Environment-Aware Flow Matching for Unified 3D Molecular Conformation Generation"
_ICLR.cc/2026/Conference — Submitted to ICLR 2026_

### Official Review · Reviewer_PeAA · 2025-10-21

**Soundness:** 2
**Presentation:** 3
**Contribution:** 2
**Rating:** 2
**Confidence:** 3

**Summary:**

This work argues that previous molecular conformation generation methods lack the ability to model environment-conditioned conformations. To address this limitation, the authors propose **EVA-Flow**, a method that generates conformations under four environmental conditions: vacuum, protein–ligand docking, solvation, and crystal packing. The overall framework integrates an **environment embedding module**, a **molecule–environment interaction encoder**, and a **flow matching decoder**.

**Strengths:**

1. The consideration of environment-conditioned molecular conformations introduces a certain level of novelty to the study.
2. The manuscript is clearly written and logically organized.
3. The overall framework, a VAE with an encoder and an FM decoder, is newly introduced in this area. The ablation study in Section 4.5 provides strong evidence supporting the effectiveness of the framework and the combined loss functions.

**Weaknesses:**

1. The encoder does not appear to be SE(3)-invariant with respect to molecular positions. In other words, the latent representation $z$ should remain unchanged when the molecular coordinates $x \in \mathbb{R}^{N\times3}$ are rotated by a rotation matrix $R \in \mathbb{R}^{3\times3}$.
2. There are no additional baselines for comparison. I believe there may exist well-established methods tailored for each environment-conditioned scenario. For example, in **Environment: Vacuum** (using the GEOM-Drugs dataset), is the setting essentially the same as in many previous works [1, 2, 3]? If there is no substantial difference, then the results reported in **Table 1 (Environment: Vacuum)** may not be particularly meaningful. The same concern applies to **Environment: Docking** (using the PDBBind-v2020 dataset) [4].
3. The “pretrain-then-finetune” paradigm has long been a standard practice in most areas. Although the experiments show that both **Pretrain + Individual** and **Pretrain + Unified** setups consistently outperform the **Unified (no pretraining)** and **Individual** baselines, I view this more as a conventional approach for improving results rather than a major contribution worth emphasizing.

[1] Xu M, Yu L, Song Y, et al. GeoDiff: A Geometric Diffusion Model for Molecular Conformation Generation[C]//International Conference on Learning Representations.

[2] Wang Y, Elhag A A A, Jaitly N, et al. Swallowing the Bitter Pill: Simplified Scalable Conformer Generation[C]//International Conference on Machine Learning. PMLR, 2024: 50400-50418.

[3] Hassan M, Shenoy N, Lee J, et al. Et-flow: Equivariant flow-matching for molecular conformer generation[J]. Advances in Neural Information Processing Systems, 2024, 37: 128798-128824.

[4] Corso G, Deng A, Polizzi N, et al. Deep Confident Steps to New Pockets: Strategies for Docking Generalization[C]//The Twelfth International Conference on Learning Representations.

**Questions:**

1. How can you demonstrate that your framework performs better than an Equivariant Diffusion or Flow Matching method [1, 3] augmented with a conditioning module (e.g., adding your environment embedding to the model inputs)? I understand that this is not a strictly fair comparison and that there are no official results available. However, considering **Weakness 2**, in **Environment: Vacuum** (GEOM-Drugs), your results are much worse than those of ET-Flow [3].
2. What is the efficacy of the method, given that the main results in Table 1 were obtained from the large-sized (68.3M) model?

---

> ### Author Response · Authors · 2025-11-22
> **Response to Reviewer PeAA (1/2)**
>
> We thank reviewer PeAA for the thoughtful feedback. EVA-Flow is designed to unify conformation generation across diverse molecular environments within a single model architecture. In response to the reviewer’s comments, we have clarified the architectural choices of EVA-Flow, and expanded baseline comparisons to better demonstrate the method’s generality and effectiveness. Below we provide detailed responses to each point.
>
> Weaknesses:
>
> 1. The encoder does not appear to be SE(3)-invariant with respect to molecular positions. In other words, the latent representation $z$ should remain unchanged when the molecular coordinates $x \in \mathbb{R}^{N\times3}$ are rotated by a rotation matrix $R \in \mathbb{R}^{3\times3}$.
>
> Response:
>
> We appreciate the reviewer’s observation. EVA-Flow adopts a hybrid design with an SE(3)-equivariant decoder and a non-equivariant encoder. We chose non-equivariance for the encoder because:
>
> a. Environment-conditioned encoder: The encoder operates in a conditional context, such as a protein binding pocket, to capture the interaction between the molecule and the environment. The absolute orientation of the molecule carries significance to capture this interaction and should not be “factored out” by an equivariant encoder.
>
> b. Empirical justification: We experimented with replacing the encoder GCNs with an EGNN architecture. However, this change did not improve conformer quality and roughly doubled the training time.
>
> This result aligns with a growing body of literature showing that non-equivariant models can perform on par with explicitly equivariant ones by learning the symmetry from data. For example, MCF (Wang et al., 2024) and AlphaFold 3 (Abramson et al., 2024) both employ diffusion-based architectures without explicitly enforcing SE(3) equivariance, yet achieve state-of-the-art results. Furthermore, Shen et al. (2025) find that a symmetry-agnostic model becomes nearly equivariant within the first 1k–10k training steps (well before final convergence). In related tasks, such as energy and force prediction, non-equivariant Transformer backbones (Levine et al. (2025), Elhag et al. (2025), and Kreiman et al. (2025)) match or surpass equivariant models on the OMol25 benchmark.
>
> In summary, EVA-Flow retains equivariance where it is essential (in the vector field decoder) and uses non-equivariant components where symmetry is not ideal (environment-conditioned encoder) or removed (base distribution). Our hybrid design achieves strong performance, while avoiding the computational overhead of full equivariance.
>
> References
>
> Y. Wang et al., “Swallowing the Bitter Pill: Simplified Scalable Conformer Generation”, ICML 2024
>
> J. Abramson et al., “Accurate structure prediction of biomolecular interactions with AlphaFold 3”, Nature 2024
>
> M. Shen et al., “Training Dynamics of Learning 3D-Rotational Equivariance”, OpenReview 2025
>
> A. Elhag et al., “Learning Inter-Atomic Potentials without Explicit Equivariance”, arXiv 2025
>
> T. Kreiman et al., “Transformers Discover Molecular Structure Without Graph Priors”, arXiv 2025
>
> D. Levine et al., “The Open Molecules 2025 (OMol25) Dataset, Evaluations, and Models”, arXiv 2025
>
> 2. There are no additional baselines for comparison. I believe there may exist well-established methods tailored for each environment-conditioned scenario. For example, in Environment: Vacuum (using the GEOM-Drugs dataset), is the setting essentially the same as in many previous works [1, 2, 3]? If there is no substantial difference, then the results reported in Table 1 (Environment: Vacuum) may not be particularly meaningful. The same concern applies to Environment: Docking (using the PDBBind-v2020 dataset) [4].
>
> Response:
>
> We agree that the Vacuum and Docking settings are closely related to prior work. In our original submission, we included comparisons to ETFlow (Vacuum) and DiffDock-L (Docking). Now we have expanded our baseline coverage:
>
> a. Vacuum (GEOM-Drugs): we now include Torsional Diffusion, GeoDiff, MCF, and GeoMol.
>
> b. Docking (PDBBind-v2020): we add DiffDock, GNINA, and TankBind.
>
> These comparisons are presented in the updated Figure 2 and Table 1. EVA-Flow achieves performance comparable to Torsional Diffusion and ET-Flow (Gaussian prior) in the Vacuum setting, and outperforms all prior methods on the Docking task.
>
> Crucially, these results are obtained using a single unified model without requiring environment-specific tuning. This highlights EVA-Flow’s ability to generalize across diverse environments while retaining competitive or superior performance compared to specialized models.

---

> ### Author Response · Authors · 2025-11-22
> **Response to Reviewer PeAA (2/2)**
>
> Weaknesses:
>
> 3. The “pretrain-then-finetune” paradigm has long been a standard practice in most areas. Although the experiments show that both Pretrain + Individual and Pretrain + Unified setups consistently outperform the Unified (no pretraining) and Individual baselines, I view this more as a conventional approach for improving results rather than a major contribution worth emphasizing.
>
> Response:
>
> We agree that the “pretrain-then-finetune” paradigm is a well-established strategy across many domains. However, it has seen limited application in molecular conformation generation. To our knowledge, EVA-Flow is the first to systematically demonstrate that pretraining on large-scale conformations, followed by unified finetuning on diverse environments, enables a single model that matches or outperforms environment-specific models. We believe this unified capability represents a meaningful step toward scalable and generalizable molecular modeling.
>
> Questions:
>
> 1. How can you demonstrate that your framework performs better than an Equivariant Diffusion or Flow Matching method [1, 3] augmented with a conditioning module (e.g., adding your environment embedding to the model inputs)? I understand that this is not a strictly fair comparison and that there are no official results available. However, considering Weakness 2, in Environment: Vacuum (GEOM-Drugs), your results are much worse than those of ET-Flow [3].
>
> Response:
>
> EVA-Flow builds directly on ETFlow with a Gaussian prior [3]. What sets EVA-Flow apart is its unified design: a single model that generalizes across diverse environments, including vacuum, protein-ligand docking, solvent, and crystal, by integrating an environment-conditioned encoder and a learned base distribution. Ablation studies show that both components are essential for cross-environment generalization.
>
> Regarding performance in the Vacuum (GEOM-Drugs) setting, EVA-Flow outperforms GeoDiff [1] and achieves results comparable to ETFlow with a Gaussian prior [3]. We acknowledge that EVA-Flow underperforms ETFlow with a harmonic prior, and we are currently working on incorporating harmonic priors into our base distribution to further improve vacuum performance.
>
> Critically, EVA-Flow is designed not to optimize for a single environment, but to perform well across all of them with one model. For example, on the Docking (PDBBind) task, the same EVA-Flow model surpasses DiffDock-L [4].
>
> We believe this unified capacity represents a meaningful advancement beyond adapting existing models with conditioning modules, and opens the door to scalable, general-purpose conformer generation.
>
> 2. What is the efficacy of the method, given that the main results in Table 1 were obtained from the large-sized (68.3M) model?
>
> Response:
>
> Our primary objective is to develop a unified model capable of generating accurate conformations across diverse environments. In this setting, we find that model capacity is a key enabler of generalization to capture complex molecule-environment interactions.
>
> The 68.3M parameter model allows EVA-Flow to match or exceed specialized models on each individual task using a single shared architecture, demonstrating that scaling contributes to performance and unification. We believe this highlights the method’s practical efficacy and scalability.

---

> ### Comment · Reviewer_PeAA · 2025-11-22
> **Response for Rebuttal**
>
> Thank you for the updates and for your hard work in addressing my comments.
>
> 1. **Response to Weakness 1**
>
>    I am indeed aware of this phenomenon in the cited works. Your empirical justification — *“We experimented with replacing the encoder GCNs with an EGNN architecture.”* — is understandable and reasonable. However, I still believe that including more detailed experimental analysis in the manuscript itself would be helpful (this is a suggestion, not a strict requirement).
>
> 2. **Response to Weakness 2 and Question 1**
>
>    Thank you for providing the new results. In Fig. 2, at the smaller parameter scale (ET-Flow (8.3M) vs. your small model (9.4M)), your method is still clearly suboptimal compared to ET-Flow (Gaussian).
>
> 3. **Response to Weakness 3**
>
>    As I mentioned in my original review, I acknowledge all your additional experiments on the “pretrain-then-finetune” setting. However, I still firmly believe that this alone does not constitute a sufficiently strong contribution at the ICLR level.

---

### Official Review · Reviewer_Vh2w · 2025-10-26

**Soundness:** 1
**Presentation:** 3
**Contribution:** 1
**Rating:** 2
**Confidence:** 4

**Summary:**

The paper proposes EVA-Flow, a unified framework for environment-aware conformation generation.

In particular, EVA-Flow couples a VAE with an environment-conditioned flow-matching decoder and a learned base distribution, thereby overcoming the limitations of existing environment-agnostic methods.

Experiments demonstrate that EVA-Flow truly generates distinct conformations for the same molecule in different environments.

**Strengths:**

1. Considering that the same molecule can adopt different conformations in different environments, it is very necessary and meaningful to construct such a unified framework for environment-aware conformation generation.

2. The paper is well-written, and the method introduction is quite detailed.

**Weaknesses:**

1. As mentioned in Section 2.2, both the Encoder and the Base Distribution Network are based on GCN, indicating that EVA-Flow lacks SE(3) equivariance—an essential property for conformation generation.

2. The paper claims to construct a unified framework for conformation generation across different environments. However, the experimental results indicate that the best performance is achieved only through fine-tuning the pre-trained model separately for each scenario, thereby failing to support the claimed contribution.

3. The four environments mentioned in this paper are already well-defined, and numerous methods have been proposed for them. However, the paper fails to compare EVA-Flow with any of these previous methods, which is clearly inappropriate.

**Questions:**

1. For Table 2, I would like to know which training strategy is used to obtain those generated conformations. Is it Pretraining + Individual Finetuning, Pretraining + Unified Finetuning, Unified Finetuning, or Individual Finetuning?

2. I would like to know whether you provide code, as this is crucial for reproducibility.

---

> ### Author Response · Authors · 2025-11-22
> **Response to Reviewer Vh2w (1/2)**
>
> We thank Reviewer Vh2w for the insightful feedback. We have clarified the architectural choices of EVA-Flow, expanded baseline comparisons, and strengthened the Pretrain+Unified results by increasing model capacity. Below we provide a point-by-point response.
>
> Weaknesses:
>
> 1. As mentioned in Section 2.2, both the Encoder and the Base Distribution Network are based on GCN, indicating that EVA-Flow lacks SE(3) equivariance—an essential property for conformation generation.
>
> Response:
>
> We agree that SE(3) equivariance is important for molecular conformation generation. However, we emphasize that both explicitly equivariant and non-equivariant modeling paradigms can achieve competitive performance in conformation generation.
>
> Our model, EVA-Flow, adopts a hybrid design. The flow-matching decoder is SE(3)-equivariant, ensuring that its generative dynamics transform correctly under rotations and translations. On the other hand, the Encoder and Base distribution use lightweight non-equivariant GCNs. This design is motivated by the following considerations:
>
> a. Environment-conditioned Encoder: The Encoder operates in a conditional context, such as a protein binding pocket, to capture the interaction between the molecule and the environment. The absolute orientation of the molecule carries significance to capture this interaction and should not be “factored out” by an equivariant encoder.
>
> b. Symmetry-normalized Base distribution: We remove global symmetry from the base distribution by center-of-mass normalization (translation) and RMSD alignment (rotation), which models the base distribution in a canonical frame. Therefore, explicit equivariance may not be necessary in this component.
>
> c. Empirical justification: We experimented with replacing the Encoder and Base GCNs with an EGNN architecture. However, this change did not improve conformer quality and roughly doubled the training time.
>
> This outcome aligns with a growing body of literature suggesting that non-equivariant models can perform on par with explicitly equivariant ones by learning the symmetry from data. For example, MCF (Wang et al., 2024) and AlphaFold 3 (Abramson et al., 2024) both employ diffusion-based architectures without explicitly enforcing SE(3) equivariance, yet achieve state-of-the-art results. Furthermore, Shen et al. (2025) find that a symmetry-agnostic model becomes nearly equivariant within the first 1k–10k training steps (well before final convergence).
> In related tasks, such as energy and force prediction, non-equivariant Transformer backbones (Levine et al. (2025), Elhag et al. (2025), and Kreiman et al. (2025)) match or surpass equivariant models on the OMol25 benchmark.
>
> In summary, EVA-Flow retains equivariance where it is essential (in the vector field decoder) and uses non-equivariant components where symmetry is either removed (Base distribution) or not ideal (environment-conditioned Encoder). Our hybrid design achieves strong performance, while avoiding the computational overhead of full equivariance.
>
> References
>
> Y. Wang et al., “Swallowing the Bitter Pill: Simplified Scalable Conformer Generation”, ICML 2024
>
> J. Abramson et al., “Accurate structure prediction of biomolecular interactions with AlphaFold 3”, Nature 2024
>
> M. Shen et al., “Training Dynamics of Learning 3D-Rotational Equivariance”, OpenReview 2025
>
> A. Elhag et al., “Learning Inter-Atomic Potentials without Explicit Equivariance”, arXiv 2025
>
> T. Kreiman et al., “Transformers Discover Molecular Structure Without Graph Priors”, arXiv 2025
>
> D. Levine et al., “The Open Molecules 2025 (OMol25) Dataset, Evaluations, and Models”, arXiv 2025

---

> ### Author Response · Authors · 2025-11-22
> **Response to Reviewer Vh2w (2/2)**
>
> Weaknesses:
>
> 2. The paper claims to construct a unified framework for conformation generation across different environments. However, the experimental results indicate that the best performance is achieved only through fine-tuning the pre-trained model separately for each scenario, thereby failing to support the claimed contribution.
>
> Response:
>
> We agree that the original Pretrain+Unified setup underperformed the Pretrain+Individual one, but this was due to architectural undercapacity rather than a limitation of the framework itself. In our updated experiments, we increased the capacity of the Environment network from two layers to five. This change led to a significant performance gain, and the Pretrain+Unified setting now outperforms Pretrain+Individual in all environments (see updated Table 1 and Figure 2). These results confirm that a single unified model can match or exceed the performance of specialized models, provided it has sufficient capacity to handle the diversity of environments.
>
> 3. The four environments mentioned in this paper are already well-defined, and numerous methods have been proposed for them. However, the paper fails to compare EVA-Flow with any of these previous methods, which is clearly inappropriate.
>
> Response:
>
> We appreciate the reviewer’s concern regarding comparisons to prior work. We originally compared EVA-Flow to ETFlow and DiffDock-L, and now we have added more comprehensive baselines: GeoDiff, Torsional Diffusion, MCF, and GeoMol for the Vacuum setting; and DiffDock, GNINA, and TankBind for the Docking task. The updated comparisons are presented in Figure 2 and Table 1.
>
> EVA-Flow achieves performance comparable to Torsional Diffusion and ETFlow (Gaussian prior) in the Vacuum setting, and outperforms all prior methods on the Docking task. These results demonstrate EVA-Flow’s ability to match or exceed specialized models across environments while maintaining a unified and generalizable architecture.
>
> Questions:
>
> 1. For Table 2, I would like to know which training strategy is used to obtain those generated conformations. Is it Pretraining + Individual Finetuning, Pretraining + Unified Finetuning, Unified Finetuning, or Individual Finetuning?
>
> Response:
>
> The results in Table 2 correspond to the Pretrain+Unified Finetuning setup. We have clarified this in the table caption and in the main text to avoid confusion.
>
>
> 2. I would like to know whether you provide code, as this is crucial for reproducibility.
>
> Response:
>
> We agree that releasing code is essential for reproducibility. We plan to release the code following required internal checks.

---

### Official Review · Reviewer_tjH8 · 2025-10-31

**Soundness:** 3
**Presentation:** 2
**Contribution:** 3
**Rating:** 6
**Confidence:** 4

**Summary:**

This paper introduces EVA-Flow, a hybrid model combining a variational autoencoder with a flow-matching decoder, designed to recover 3D molecular conformations from 2D molecular graphs. Existing methods often overlook the critical dependence of molecular conformation on the surrounding environment (e.g., crystal lattices, protein binding pockets), limiting their generalizability. EVA-Flow addresses this limitation through two core innovations: (1) It integrates environmental information as a conditional embedding within a flow-matching decoder, endowing the model with environment-aware generation capabilities; (2) It employs a graph convolutional network-based encoder to fuse molecular and environmental information into a shared latent space, effectively capturing their feature relationships.
Experimental results demonstrate that EVA-Flow achieves superior performance on both Vacuum and Protein-Ligand Docking datasets compared to ETFlow and DiffDock. The model's performance is further enhanced through pre-training followed by unified fine-tuning. Visualization analyses confirm that EVA-Flow can generate distinct conformations for the same molecule in different environments, validating its robust environment-aware capabilities and generalizability.

**Strengths:**

1. The work is innovative in its explicit integration of environmental information into conformation generation, proposing a novel hybrid architecture based on a variational autoencoder and flow matching.
2. The model design is sound, utilizing a graph convolutional network to effectively fuse molecular and environmental features within a latent space while preserving their structural relationships.
3. The comprehensive experimental validation across multiple datasets, supplemented by visualization analysis, lends credibility to the reported results.
4. This research provides a more generalizable tool for molecular conformation generation, with significant potential for practical applications in drug and crystal design.

**Weaknesses:**

1. The experimental configuration is suboptimal. The "Experiments" section lacks direct comparisons with key contemporary models. Using DiffDock and ETFlow merely as reference points, rather than in a rigorous, head-to-head benchmark, makes it difficult to accurately assess EVA-Flow's performance advantages.
2. The model diagram (Figure 1) is insufficiently detailed. While it illustrates the overall EVA-Flow architecture, it fails to clearly depict how the four loss functions (Equations 8-11) relate to specific model components, hindering understanding of the training process.
3. The conclusion is underdeveloped. It primarily reiterates the main contributions and findings without a critical discussion of the study's limitations or a clear, actionable outlook for future work.
4. The placement and content of the related work section are problematic. Positioning it after the methodology section ("Environment-Aware Flow Matching") deviates from standard academic structure. Furthermore, the section itself does not adequately articulate the primary motivation for EVA-Flow or the core challenges and innovations in its design.

**Questions:**

1. The "Experiments" section should be revised to include direct, comprehensive comparisons with state-of-the-art models to more convincingly demonstrate EVA-Flow's advantages.
2. Figure 1 should be revised to explicitly include and annotate the loss terms, making the training mechanism more transparent.
3. The "Conclusion" should be expanded to include a summary of the study's limitations and to propose specific, feasible directions for future research.
4. The "Related Work" section should be relocated to follow the "Introduction" and precede the methodology section. Its content should be revised to clearly establish the research gap, the motivation for EVA-Flow, and a detailed discussion of the core challenges and how the proposed innovations address them.

---

> ### Author Response · Authors · 2025-11-22
> **Response to Reviewer tjH8**
>
> We thank Reviewer tjH8 for the constructive feedback. We have expanded baseline comparisons, added loss calculation to the model diagram, added limitations and future directions to the Conclusion, and relocated and revised the Related Work section. Below we provide a point-by-point response.
>
> Weaknesses
>
> 1. The experimental configuration is suboptimal. The "Experiments" section lacks direct comparisons with key contemporary models. Using DiffDock and ETFlow merely as reference points, rather than in a rigorous, head-to-head benchmark, makes it difficult to accurately assess EVA-Flow's performance advantages.
>
> Response:
>
> We have added more comparisons with recent work on Vacuum and Docking environments. Specifically, we have included ETFlow, GeoDiff, Torsional Diffusion, MCF, and GeoMol for Vacuum, and DiffDock, DiffDock-L, GNINA, and TankBind for Docking. These comparisons are presented in updated Table 1 and Figure 2.
>
> EVA-Flow achieves performance comparable to Torsional Diffusion and ETFlow (Gaussian prior) in the Vacuum setting, and outperforms all prior methods on the Docking task. These results demonstrate EVA-Flow’s ability to match or exceed specialized models across environments while maintaining a unified and generalizable architecture.
>
> 2. The model diagram (Figure 1) is insufficiently detailed. While it illustrates the overall EVA-Flow architecture, it fails to clearly depict how the four loss functions (Equations 8-11) relate to specific model components, hindering understanding of the training process.
>
> Response:
>
> We have revised Figure 1 to add the loss terms with direct and clear identification of how the model components relate to them.
>
> 3. The conclusion is underdeveloped. It primarily reiterates the main contributions and findings without a critical discussion of the study's limitations or a clear, actionable outlook for future work.
>
> Response:
>
> We thank the reviewer for the suggestion. We originally placed the Limitation section in Appendix H due to page limits. We have moved it to the Conclusion.
>
> “While EVA-Flow demonstrates the feasibility of unified, environment-aware conformation generation, several limitations remain and point to valuable future directions. First, we focus on a specific architecture: a single FM decoder with GCN-based networks for the base, encoder, and environment modules. More expressive architectures may further improve generalization and accuracy. Second, our base distribution assumes a simple Gaussian prior; extending it to more structured priors such as harmonic prior could better capture the geometry of conformers. Third, while EVA-Flow is designed to scale, our current models are of modest sizes. Increasing the capacity of the environment encoder and latent embeddings could further enhance the model's ability to capture complex molecule-environment interactions. Finally, our evaluation was limited to four environments with high-quality datasets. Extending EVA-Flow to more diverse and challenging contexts, such as surface adsorption on catalysts or solid-liquid interfaces, will require new benchmarks with carefully curated data and poses an exciting direction for future work.”
>
> 4. The placement and content of the related work section are problematic. Positioning it after the methodology section ("Environment-Aware Flow Matching") deviates from standard academic structure. Furthermore, the section itself does not adequately articulate the primary motivation for EVA-Flow or the core challenges and innovations in its design.
>
> Response:
>
> We have moved the related work section to appear before the methodology section. We have also revised the related work section to better describe the motivation, challenges, and innovations.
>
> “Key challenges in this space include: (i) the lack of unified models that generalize across diverse environments, and (ii) the difficulty of incorporating complex conditioning inputs such as structured environments into generative frameworks. EVA-Flow addresses these challenges by combining an environment-aware encoder with a FM decoder, enabling a single model to generate accurate conformers across different environments without requiring task-specific pipelines.”
>
> Questions
>
> 1. Response: We have added direct, comprehensive comparisons with state-of-the-art models that are presented in updated Table 1 and Figure 2. EVA-Flow achieves performance comparable to Torsional Diffusion and ETFlow (Gaussian prior) in the Vacuum setting, and outperforms all prior methods on the Docking task.
>
> 2. Response: We have revised Figure 1 to add the loss terms with direct and clear identification of how the model components relate to them.
>
> 3. Response: We have moved limitations and future plans to the Conclusion.
>
> 4. Response: We have relocated and revised the Related Work to follow the Introduction and explicitly discussed the challenges in unified conformation generation and EVA-Flow's motivations and innovations.

---

### Author Response · Authors · 2025-12-03
**Summary of Key Revisions and Responses to Reviewer Feedback**

We thank all reviewers for their thoughtful feedback. Below we summarize the core contributions of the paper, the major concerns raised across reviews, and the concrete revisions and new experiments added in the updated manuscript.

Summary of Contributions

1. Unified, environment-aware flow matching framework for molecular conformation generation across four diverse environments (vacuum, protein-ligand docking, solvent, and crystal) using a single model architecture.

2. Large-scale pretraining + unified finetuning enables a single model to match or outperform environment-specific models.

3. Comprehensive analysis demonstrates that EVA-Flow meaningfully incorporates environment information.

Major Cross-Reviewer Concern #1: “Pretrain+Unified underperforms Pretrain+Individual, weakening the unified framework claim.”

Response & Revision:

We identified under-capacity in the environment network as the root cause. Increasing the environment network from 2 to 5 layers substantially improved unified finetuning.

Updated results (Table 1, Fig. 2) now show Pretrain+Unified outperforming Pretrain+Individual across all environments, confirming that a single sufficiently expressive model can match or exceed specialized models.

Major Cross-Reviewer Concern #2: “Encoder and Base networks are not SE(3)-equivariant.”

Response & Rationale:

EVA-Flow intentionally adopts a hybrid design:

1. Environment-conditioned encoder: Absolute orientation is meaningful for modeling molecule-environment interactions. The encoder receives two inputs: the molecule’s ground-truth conformation and an invariant environment embedding produced by the environment network. Because the environment embedding does not transform under SE(3), enforcing equivariance in the encoder would impose a mathematical contradiction: an equivariant encoder would be required to ignore absolute molecule orientation, yet meaningful molecule-environment interactions depend explicitly on that orientation.

2. Base distribution: We remove global symmetry using center-of-mass normalization and RMSD alignment, effectively placing samples in a canonical frame where explicit equivariance is unnecessary.

3. Empirical evidence: Replacing Encoder/Base GCNs with EGNNs did not improve accuracy and doubled training time.

4. Literature alignment: State-of-the-art systems such as AlphaFold3, MCF, and OMol25 baselines also rely on non-equivariant architectures while achieving excellent geometric fidelity.

Together, these results support the hybrid approach and clarify that EVA-Flow retains equivariance where essential (decoder) while avoiding unnecessary overhead.

Major Cross-Reviewer Concern #3: "Limited baseline comparison to the literature."

We added all major contemporary baselines:

1. Vacuum (GEOM-Drugs): GeoDiff, Torsional Diffusion, MCF, GeoMol, ETFlow

2. Docking (PDBBind): DiffDock, DiffDock-L, GNINA, TankBind

Key findings: EVA-Flow is comparable to ETFlow (Gaussian prior) and Torsional Diffusion in Vacuum. EVA-Flow outperforms all prior methods on Docking. Crucially, these results come from a single unified model, without any environment-specific pipelines.

Other Revisions Requested by Reviewers:

1. Clarified training strategies (e.g., Table 2 uses Pretrain+Unified).

2. Expanded model diagram to explicitly show how each loss term connects to model components.

3. Moved and strengthened Limitations section with concrete future directions.

4. Revised Related Work to better motivate the unified setting and articulate the main challenges.

5. Confirmed code release plan following internal checks.

Summary

By improving unified model performance, clarifying the architectural motivation for the hybrid equivariant/non-equivariant design, expanding baselines, and revising the manuscript for clarity, we believe the updated paper fully addresses the reviewers’ concerns and significantly strengthens the submission.

---

### Meta-Review · Area_Chair_noP6 · 2026-01-09

**Summary:**

The paper proposes "EVA-Flow," a unified framework for environment-aware 3D molecule generation. While the model aims to handle multiple environments in a single model , it suffers from deficiencies in experimental rigor and theoretical soundness. Primary concerns include a performance gap in vacuum generation compared to SOTA methods , a flawed docking evaluation protocol, and an architectural choice (non-equivariant encoder with an equivariant decoder) that is theoretically inconsistent

**Reviewer Concerns:**

•	Addressed: The authors expanded baseline comparisons and addressed the "Unified" training benefit by increasing environment network capacity to show improvements over individual fine-tuning. Manuscript structure and clarity were also improved.

•	Outstanding: The model's vacuum performance remains significantly inferior to SOTA and its direct baseline ETFlow. The docking evaluation is still unconvincing as it lacks standard benchmarks. the theoretical inconsistency regarding the non-equivariant encoder and its inability to effectively pass orientation information to the equivariant decoder remains  unresolved.

**Reviewer Scores:**

•	Reviewer PeAA (2) and Vh2w (2): Likely to maintain their original low scores as the rebuttal failed to resolve fundamental concerns about evaluation validity and theoretical design.

•	Reviewer tjH8 (6): Likely to lower their score given the critical deficiencies identified during the discussion regarding performance degradation and theoretical flaws.

---

### Decision · Program_Chairs · 2026-01-26

Reject